# Developing a Virtual Museum: Experience from the Design and Creation Process

**Felipe Besoain** [1,*] , **Liza Jego** [1] and **Ismael Gallardo** [2]

1   School of Videogames Development and Virtual Reality Engineering, Faculty of Engineering, Campus Talca, Universidad de Talca, Talca 3460000, Chile; ljego@utalca.cl
2   Faculty of Psychology, Campus Talca, Universidad de Talca, Talca 3460000, Chile; igallardo@utalca.cl
*   Correspondence: fbesoain@utalca.cl

**Abstract:** Virtual reality technology has grown significantly in recent years. The arrival of Head Mounted Displays (HDM) on the market for end-users has positioned these technologies as a new channel to promote new simulated or contextualized experiences. We have used the design and creation strategy to develop a virtual reality experience for the Oculus GO and Quest HDM. We digitized 30 pieces from nine local museums to provide an experience guided by a character that represents the main artisan work of the local region. A usability test was performed, showing that participants felt a high degree of immersion and realism. They were able to complete the assigned tasks, and results suggest that the software meets the main objective. Furthermore, the creation of this virtual reality (VR) experience has shown how important it is to make users a part of the creation process, as well as to develop a process to make the software useful to them and other users. Some recommendations are made based on the experience of the development, and comments are given on each step of the design and creation strategy.

**Keywords:** virtual museum; cultural heritage; virtual environments; interaction techniques; software development

## 1. Introduction

Several advances have been made in the Information and Communication Technologies (ICT) that allow the deployment and distribution of information in multiple or different devices. Access to the Internet and the mass use of it through mobile devices and PC technology has become very attractive for cultural and archival institutions to interact and show content to the community through digital channels [1]. Before, the only way to appreciate these tangible heritages was by visiting the museums and other cultural institutions in person.

Virtual reality (VR) technology has become important for providing users with unique experiences due to the high sense of immersion that increases their perception and, in some cases, provokes the sense of being there (presence) [2,3]. It is important to note that the experience of appreciating art in person is irreplaceable. However, this technology allows the creation of contextualized virtual environments that focus on users' experiences. Some studies have even shown that an immersive visualization environment can help enhance learning [4].

VR technologies are becoming more widespread and popular. The market for VR software and hardware is expected to grow rapidly in the next few years, with an estimated growth from 2019 to 2022 of 6.2 billion dollars (USD) to more than 16 billion (USD). The countries that spend the most on VR technologies are the United States and China [5].

This technology has several applications in different fields such as training [6], applications for people with special needs [7], education [8], and tourism [9,10], among others.

In particular, VR technology can be used in museums, with the creation of virtual products. An example of a virtual product is the display of objects through the

use of virtual or augmented reality. In this way, the use of VR in museums modifies the way users interact with museum content, bringing the content closer to them [11]. Virtual products are regularly used in a variety of ways in the areas of tourism and cultural heritage, for example, for research, reconstruction, preservation, documentation and dissemination of cultural heritage. Thus, the use of VR technologies and virtual products can be considered an important tool for promoting and conserving cultural heritage [12]. Using technology to preserve cultural or natural heritage is known as digital heritage [13], which also includes various technological products that are used in computer-based applications, including software, text, audio, images, etc. In addition, in the use of serious games, augmented reality has been shown to engage and motivate participants to learn about cultural heritage [14].

The use of virtual products and digital heritage is also relevant to the idea of a virtual museum, a concept that has existed since even before widespread access to the Internet [15]. The growth of VR technologies, however, opens even more opportunities for virtual museums. For example, objects from multiple collections and even multiple museums can be combined into a virtual experience, without the limitations of physical infrastructure, thus giving users greater access to museum content [16]. Moreover, the use of these technologies has shown that an immersive VR environment relates to the user's motivation to physically visit a museum [17].

In this context, users can see and experience different aspects, including, a real sensation of perspective in a 3D environment, depth for interacting with their hands, and a first-person field of view, among others. These aspects enhance the experience inside of a virtual world.

Therefore, it is relevant to explore the use of this technology in the creation of experiences that show, in a creative and unique way for users, the visualization of tangible aspects, such as the 3D museum pieces, and intangible aspects, such as cultural background.

Thus, the question to be addressed in this work is: how can we use this technology to create an experience of a virtual museum for VR Head Mounted Displays (HDM) with a central narrative that invites users to know, visit and live the experience? To answer this question, we propose the use of VR technology and develop an experience for Oculus GO [18] and Quest [19] HDM of a virtual museum with a 3D environment, interaction with 3D objects (local museum pieces), gamification, and contextualized information.

With all this comes a need to consider a wide audience so that the experience also serves as a tool to invite potential visitors to learn more about the local museum.

We used Unity 3D game development software [20] for the development of scalable and modular software. This software should meet five important criteria:

- Ease of use;
- Consideration of virtual reality motion sickness;
- Implementation of interaction mechanics with the 3D historical pieces;
- Integration of different pieces through a central narrative;
- Scalability of the experience.

The present paper is structured as follows: first, the design and creation strategy is explained with its five steps, introducing the materials and methods used to create the VR experience, including an explanation of the use cases of the application, design process, digitization of 3D pieces, and usability test; second, we present the results, which are principally the main features of the software; and finally, we describe our conclusions and future work, including some recommendations based on the experience of the development and comments on each step of the design and creation strategy.

## 2. Methodology

The methods used in this research follow the design and creation approach [21] to create a VR experience and the software that runs it. The design and creation research strategy focuses on learning through making. It is an iterative process with five steps: awareness, suggestion, development, evaluation and conclusion [21,22]. A brief descrip-

tion of each step is given, and then the rest of the methodology section is structured around each of these steps, explaining the use of this strategy in the development of the virtual museum.

In the first step, awareness, a problem is defined, which can come from literature, field research, new technological developments, or from practitioners that express a need for something. In the case of the virtual museum in the present paper, the problem came from the need of local museums to attract more potential visitors and a younger generation. This goes hand in hand with the arrival of stand alone HDM virtual reality technology to the end-user market.

In the second step, suggestion, a tentative idea is created for how the problem can be resolved. For the present work, this step included use cases as part of an iterative process to create a design idea for the virtual museum. We describe two general use cases with their key components and main functionalities. It also consisted of the creation of a central narrative to guide the VR experience. A short explanation is also given of the design of the main character in this narrative, a tutorial scene, and the user interface and environment. In this methodology section, the first two steps (awareness and suggestion) are combined due to the overlap in these processes.

Development, the third step, consists of implementing the design idea. For the current virtual museum, this step included the creation of software that responded to the needs of local museums and incorporated the new developments in virtual reality technology. In particular, the digitization and optimization of the 3D pieces are explained, along with their mechanics in a contextual graphic interface. It is important to note that the results of the implementation of this design idea are not addressed in this methodology section of the paper but rather in the following results and conclusion section. The methodology section instead presents a sort of guideline for the use of the design and creation strategy in the development step.

In the fourth step, evaluation, the developed artifact is examined, its contributions are assessed, and any unexpected results are explained. In the case of the current paper, usability testing and analysis were carried out, a description of which is offered below, including the participants and procedure. The usability test performed included a VR motion sickness questionnaire [23], with 33 participants. Once again, the results of the evaluation are presented in the results and conclusions section, while this methodology section focuses on guidelines of what was done for the evaluation.

The fifth and final step, the conclusion, collects and summarizes the design process results, identifying what has been learned and what still needs to be examined, such as deviations from expectations that could be explored in future work. These conclusions will be addressed in the final sections of the paper, including the discussion section.

### 2.1. Awareness and Suggestion

#### 2.1.1. Use Cases

We present two general use cases as an example: (1) introduction to the experience and context and (2) interaction with 3D pieces. The use cases help to understand the behavior of the software and its interactions with external actors in certain situations. The first use case establishes what happens when the software is opened for the first time (the first run of the VR experience), putting the user in the context and also introducing the VR controls and Graphical User Interface (GUI). The second presents the interaction of users with the 3D pieces and functions related to zoom in and out and movement in 360 degrees, among others.

Use case: *Introduction to the experience and context*

Actor(s): User and Controller
Purpose: Introduce the user to the experience and how to interact with it.
Summary: This use case begins when the user runs the experience on the Oculus HDM. The software starts with a tutorial scene with an introduction to the VR experience and context.

Preconditions: The user has run the VR experience in the Oculus HDM.

Use case: *Interaction with 3D pieces*

Actor(s): User and Controller
Purpose: Present to the user, through the GUI, a 3D piece with associated information.
Summary: This use case begins when the user selects a collection and then a 3D piece to see further details. The system will present the object with its information, description and other details.
Preconditions: The user has already completed the tutorial scene and knows how to interact with the GUI of the VR experience.

More details on use cases: (1) introduction to the experience and (2) context and interaction with 3D pieces, are provided in Tables 1 and 2, respectively.

**Table 1.** This table presents an extension of the *Introduction to the experience and context* use case in a conversational format, which emphasizes the interaction between the actors and the system.

| Actor's Actions | System's Answers |
| --- | --- |
| 1. This use case begins when the user runs the experience on the Oculus HDM. | 2. An orb with particles is presented to draw the user's attention. |
| 3. The user points at the orb with the controller and gets visual feedback. | 4. The orb moves in the environment to a different position. |
| 5. The user points at the orb with the controller and gets visual feedback. | 6. A main character is presented (woken up). |
| **Alternative flow** | 3. If the user does not point and interact with the controller and the orb, and instead looks somewhere else, then a notification audio sound will be played to draw the user's attention. |

**Table 2.** This table presents an extension of the *Interaction with 3D pieces* use case, in a conversational format, which emphasizes the interaction between the actors and the system.

| Actor's Actions | System's Answers |
| --- | --- |
| 1. This use case begins when the user selects a collection on the panel. | 2. The system presents a list of pieces with 2D images. |
| 3. The user selects a pieces | 4. An interface is presented with the 3D piece and its information on the right side. |
| 5. The user selects to zoom in or out. | 6. The object zooms in or out according to the previous event. |
| 7. The user selects auto spin. | 8. The object spins on the y axis on its central pivot point. |
| **Alternative flow**: | The user goes back to the previous information panel. In this case, the system will present the previous panel, i.e., Collection information. |

As a result of these use cases, there are several processes and components to consider in the implementation of the software, which requires consideration of the following aspects:

- Design of the tutorial scene;
- Design of the character with cultural value;
- Design of the GUI and environment;

- Digitization of 3D pieces and optimization;
- Mechanics for 3D pieces in a contextual GUI.

In the following subsection, we describe the central narrative that was chosen to guide the VR experience, along with the descriptions of the main aspects considered.

### 2.1.2. Narrative Summary

In our VR experience, we want to show 30 pieces from 9 different museums of the Maule region in Chile. The pieces correspond to different contexts but have been grouped according to the period of time from which they come in three different collections. This process was carried out with the participation of museum directors and a local anthropologist.

The VR experience should provide an interactive experience but also promote and reference where the pieces are located to encourage users to visit them.

In this context, we considered relevant the creation of a character with the role of a host that guides the user into the experience. This character has to be related to the region. In the search of this key element, after a creative brainstorming process, it was decided to focus on the artisans of the region and their work. The Maule region is known for its artisan pieces in greda (clay), leather and crin (horsehair weaving) [24]. The last one, crin in the city of Rari, has been recognized by the World Crafts Council as an important cultural element, naming Rari as a World Craft City for horse hair work [25].

Therefore, we took crin as a reference to create a character called Maulina, who was created by artisans and guides the user into the main experience. In this scenario, Maulina presents the environment, the main functionalities and also guides the user through the available collections.

### 2.1.3. Design of the Tutorial Scene

Because VR HDMs, like Oculus, are relatively new to the community, many people do not know how to operate these kinds of devices. Thus, it is necessary to provide a tutorial scene to make the operation of the device easier and therefore reduce operational problems in the VR experience.

The VR experience was planned to be shown in: (1) areas open to the public, such as museums, public libraries, or fairs; and (2) private spaces for users who have VR headsets.

For those reasons, Oculus Go was selected as the VR device because it is a less expensive HDM and therefore it could be easier to find more users on that platform. However, Oculus Go was discontinued [26] and was replaced by Oculus Quest [19]. So, the VR experience can be run on both devices.

The Oculus Go headset uses a 3 Degrees of Freedom (DOF) tracker for the user's head. As a result, developers recommend that for this device the experience should be: (1) seated or (2) stationary play. For seated play, the user will be seated in a chair, and for stationary play the user should not be required to move beyond reaching with arms or leaning from the torso [27].

Therefore, considering the main narrative and Maulina (the crin doll character), a concept was designed for the tutorial that allows the introduction of Maulina and at the same time shows the user how to use the controller to move their head, hands, use action buttons, etc., to enjoy a 3D experience.

The first scene, the tutorial, starts with a fully black screen. Then, particles with low illumination and sound slowly and softly appear to the right of the field of view (FOV) to guide users to naturally move their head to the right. After that, more lights and sounds appear at different points in the area to show the user that a 360° environment is there, and also, some interactions can be started with the head and controller, see Figure 1.

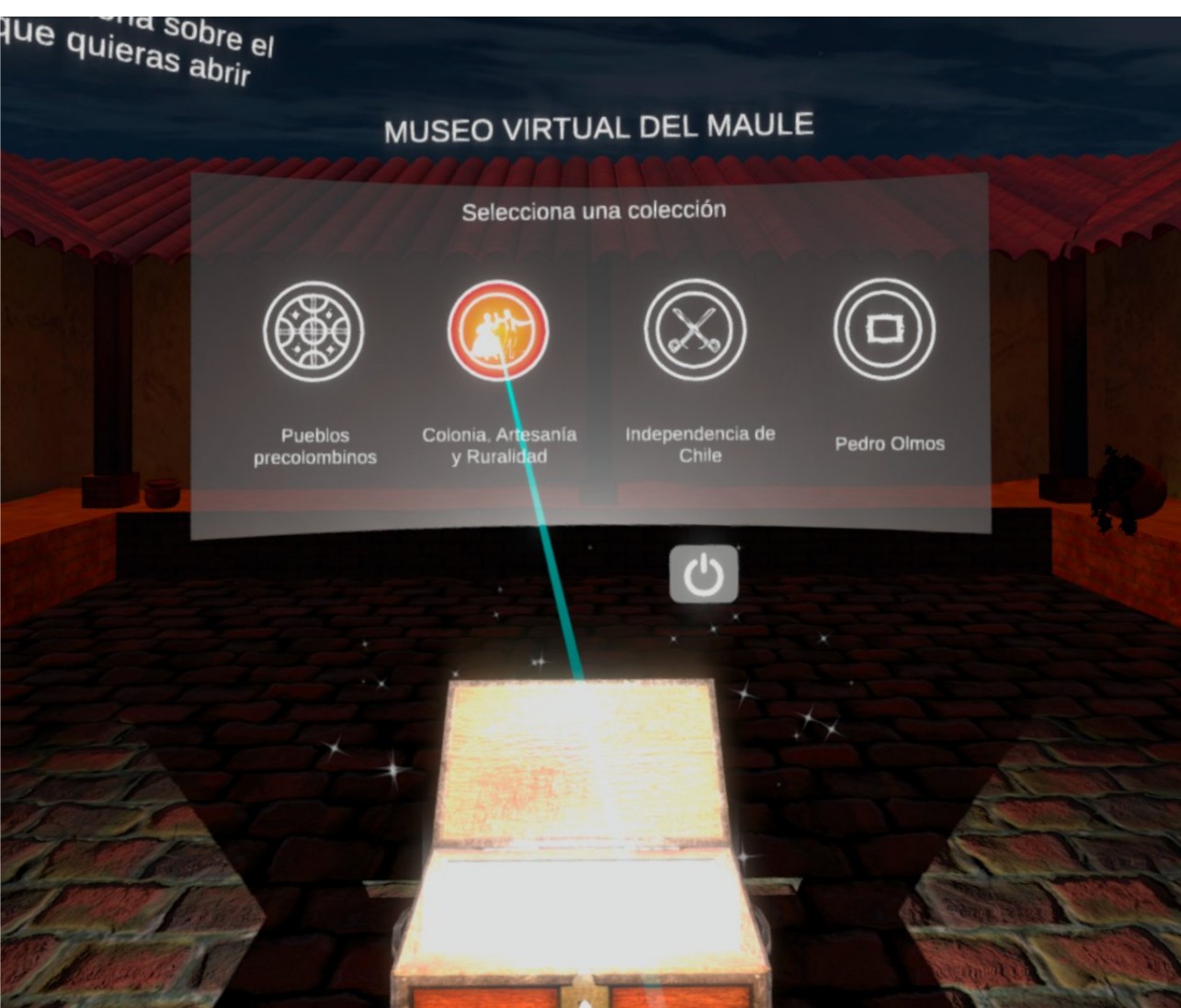

**Figure 1.** Graphical user interface of the virtual museum: the front panel where the user can access the 3D pieces through the available collections. The main GUI has three panels: front (in the image), left, and right.

### 2.1.4. Design of the GUI and Environment

When designing interactive systems like the virtual museum, it is important to give the user an experience that is enjoyable, easy, and effective to use. For that purpose, the basic guidelines of interaction design were followed, which involves identifying user needs, establishing requirements, developing design alternatives that meet those requirements, building interactive prototypes and evaluating the design throughout the process [28].

The user interface (UI) is the point of action where a user interacts with a product, machine, or systems of all kinds. The objective of the UI is to allow the operation and control of the system/product in an easy, simple, and effective way. The UI plays an important role in the creation of a friendly visual environment between the user and the technology; therefore, for this version of the VR experience, appropriate and adequate interfaces were designed for this platform in order to provide a satisfactory user experience. The ease with which the user moves between the real world and the virtual world depends on the way the information is delivered to the user; hence, when designing the interface, it was necessary to consider the environment in which the experience would take place, the desired content to display, and the graphic style that would be used to display that content.

The interface design is one of the most challenging aspects in the development of a virtual experience, considering that the virtual environment (VE) is a form of human–

computer interaction itself. In a VR system, the user interacts with helmets, glasses, gloves, and controls, among others.

The interface design of the virtual museum can be considered minimalist, as it uses thick and simple shapes and lines—a neutral design that works well in virtual reality environments. Details will be specified below.

The user interaction with the system is done through the HDM and a controller with a pointer, with which users can select options and explore the environment. To ensure that the virtual environment was efficient, effective, and satisfying to use, the interactions were designed so the user can explore easily and intuitively. In addition, it was considered that the visual representations of the virtual environment should correspond to the user's usual perception in a real environment. After several brainstorming sessions, it was defined that the user will interact with the VE through a system of curved panels that will surround the user in order to make them feel immersed in the virtual space. This "closed virtual space" is intended to establish a moderate limitation of movement in virtual space in order to make the user feel safe and prevent them from becoming disoriented (especially taking into account that these devices are new for the target audience in the local region). It is also important to emphasize that the curved menus that surround the user in a virtual space in three dimensions (3D) allow a better visualization of images and text reading [29]. In the panels, users will be able to find all the contents, collections of pieces and any other related information that can be explored by the user, see Figure 1.

The collections of pieces will be displayed on the panels with a carousel-like menu, where the user can select and inspect each piece, zoom in and view its details, as well as access the technical information of the piece. For the creation of the museum stage, elements and colors characteristic of the colonial style were used, such as lanterns, tiles, wood, and cobblestones, among others.

The use of the controller allows the user to interact with the information in the panels, so the object can be selected, manipulated, and rotated if desired. This works through a laser system that allows the user to manipulate the controller as if they were manipulating the object, giving a sense of realism to the experience.

The interfaces were tested and evaluated throughout the development of the software by a group of users, which meant a continuous process of iterations in the design to obtain the best possible interface for the VR experience.

### 2.2. Development

### 2.2.1. Digitization of 3D Pieces and Optimization

Digitization is one of the main processes to address in this work. In order to show some 3D pieces in the VR experience, it is necessary to bring physical pieces into a 3D world environment. To achieve this, a process of digitization is needed. The digitization process is defined as the conversion of analog material into a digital form that results in a digital copy available for use in computer-based applications [30].

Several technologies can be used in order to digitize objects. In this work, mainly two were used: (1) 3D scanning, see Figure 2a; and (2) photogrammetry for deriving precise and reliable 3D measurements through images [31,32], see Figure 2b. In some cases, a combination of these processes was used, mixing both technologies [33].

For 3D scanning, EinScan Pro was used, including a 3D scanner with a tripod, rotating stand and color pack. For photogrammetry, a Réflex Cámera Canon T6s EOS Rebel with lenses and white photographic curtain was used.

Both technologies result in a high fidelity product and a high image resolution, which results in a suitable model for heritage conservation and digital restoration. The laser scanner and photogrammetry provide dense point clouds. Compared to the scanner, photogrammetry is cheaper because, in order to perform this procedure, only cameras and other related accessories are needed for taking the pictures, along with alignment software to extract the 3D model.

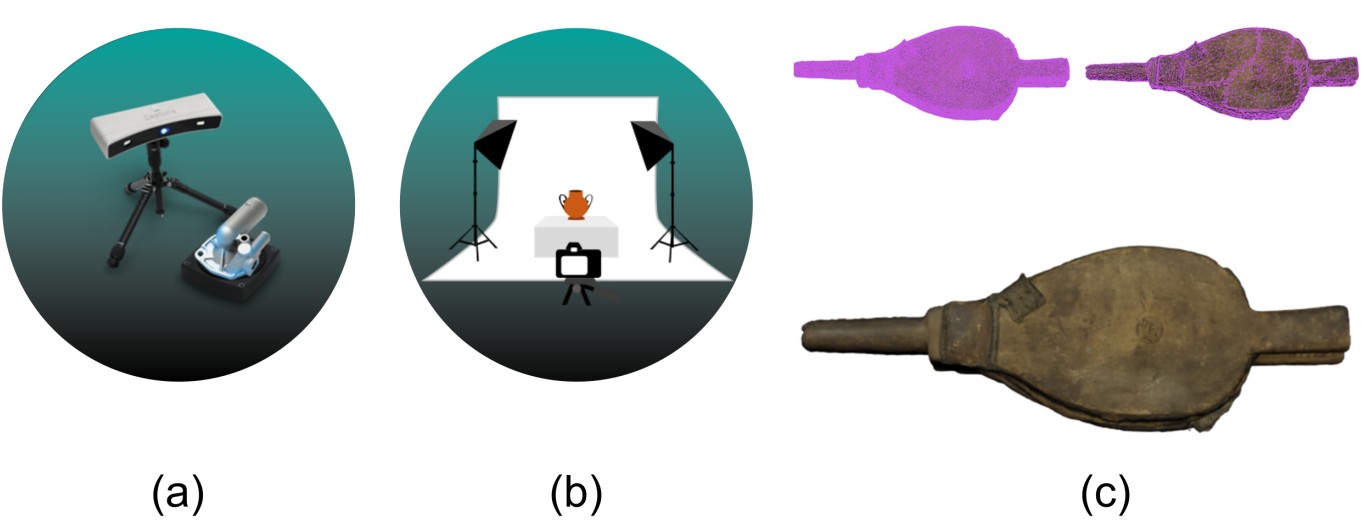

**Figure 2.** Digitization process: (**a**) 3D scanner technology; (**b**) photogrammetry technology; (**c**) optimization process of the three-dimensional model of a bellows using Autodesk 3ds Max software.

In fact, the photogrammetry process offers a cheap, flexible and accurate solution to obtain 3D point clouds and textured models. The main advantage of this photogrammetric methodology is that a cloud of points and a precise mesh object with texture can be obtained at the same time [12].

The way an object is digitized (taking an object from the real world to a 3D world) depends fully on the characteristics of the object. For example, size and the type of material are especially important to determine which technique will be used. Further information on the framework and process can be found in [34].

After the digitization, the main challenge of this process is to find a balance between the quality of the representation and the number of polygons of the 3D piece, see Figure 2c. Optimization is a key process for running the experience smoothly on the device. If it has a large number of polygons (high poly), more CPU and RAM are needed so that the frames per second (FPS) of the experience do not decrease. To digitize the objects with photogrammetry technology, the Agisoft Metashape was used. With this software, the photogrammetric processing of the digital images of the objects previously taken is carried out. With this done, the mesh optimization process of the models is carried out with the Autodesk 3ds Max software. With the use of this software in conjunction with the proper optimization process, a realistic appearance of objects was achieved, without the need to obtain a mesh of millions of polygons as the same object that is being digitized. To achieve a surface that is as similar as possible to the real object, images of objects textures can be used on the "low poly" mesh (<25,000 polygons), which allows the models to be visualized in high quality with a reduced number of polygons. The number of polygons needed and the resolution will vary according to the object that is being optimized and according to the requirements of the platform where it will be processed. More details about the model optimization process can be found in [34].

2.2.2. Mechanics for 3D Pieces in a Contextual GUI

The first version of the VR experience was made without a controller. In this version, all the interactions were controlled by a raycast. This technique is usually used when an HDM does not have any controller. In this way, the user can control and interact with the software, moving their head and leaving the pointer (with the raycast) for a certain number of seconds in a determined place.

This decision was made in order to have a version that could be used on mobile devices with cheaper converters for a VR experience, such as Google Cardboard and Google Dream.

However, it is important to note that in order to run the experience smoothly, the mobile device has to at least match the hardware features of Oculus GO. After some usability tests, users mentioned that they would love more interaction with their hands, so that they do not always need to move their head to start an interaction.

Therefore, we found that, although the experience works well with the previously described motion control, it was attractive to use the Oculus GO controller. The use of the controller gives the user a different experience because seeing in 360 degrees is separated from how the user controls the interaction with the VR environment. Use of the controller allows the user to touch something in their hand to control the experience, without the use of the raycast.

The characteristics of the controller (aspects available for programming) are buttons, trackpad and gyroscope.

In this context, we wanted to implement a special feature for the visualization of the 3D pieces. Thus, when users open a 3D piece of their interest, it has functions such as zoom in and zoom out. In that GUI, the 3D piece moves to mirror the user's motions of the controller. This means that the controller's gyroscope is used to move the 3D piece, making users feel that they have the piece in their hand.

It is known that this, in some cases, can be unreal because of the size and weight of the 3D pieces. However, it still offers the user a more engrossing interaction.

*2.3. Evaluation*

2.3.1. Usability Testing

The main purpose of the usability testing is to evaluate a product or service with real users in order to uncover existing usability problems. Usability refers to the ease with which users use software, devices, products or systems, among others. It is mainly about the effectiveness, efficiency and user satisfaction [35].

In a usability test session, participants are asked to use the product or service under study to perform specific tasks, in order to identify problems associated with user interaction with the product interface [28]. It is considered that the best way to obtain a good interaction design is to test it with real users [36]. A qualitative usability study was carried out, which consisted of collecting information on how people use the product, as it is considered the most suitable for discovering problems in the user experience [36].

The aim of this activity was to analyze the usability of the software to find significant design errors that could cause confusion or frustration in its use and that could be improved in a future version. For this purpose, usability testing sessions were carried out with real users, the results of which were analyzed later.

The results of the usability test will consist of a list of errors or problems that must be improved in the virtual reality software in addition to the collection of general opinions of the participants.

The study took place in Mauletec Virtual Reality Laboratory, where participants were instructed about the task they had to complete.

2.3.2. Participants

For usability testing, it is recommended to do the test with 5 participants because with that number it is possible to find the majority of the most common problems or errors of the product, although the number of participants required will vary according to the type of study [36]. In this case, there was a higher number of participants than recommended because there was an open call to participate.

A total of thirty-three participants (4 female, 29 male) were recruited to take part in this study, using a convenience sample. All participants were undergraduate students from Universidad de Talca aged between 18 and 27 years old. No participants had previous experience with virtual environments or virtual reality games.



### 2.3.3. Procedure

Before the usability testing session started, participants were instructed about the task they had to perform, the procedure, and how to use the Oculus GO headset and controller. Each participant was able to take the time they wanted to complete the activity. In order to be able to evaluate the ease of use and how participants learned to use the software, they were not specifically taught how to move and interact within the virtual environment. Users were asked to complete the experience sitting in a chair to reduce the motion sickness, although they could stand up if they wanted to. Average usage time of the software was 30 min. To conduct this research, an exploratory design was adopted.

Two researchers observed the behavior and gestures of the participants, and noted the problems encountered as the participants completed the task. In addition, while the participants explored the virtual environment, they were asked to comment aloud [37] on their positive and negative thoughts and observations, as well as ask questions.

Once the task was completed, participants were asked to complete the Witmer and Singer 24-question Presence Questionnaire [38], where they evaluated their sense of presence (or immersion) experienced when using the virtual reality HMD.

At the end, participants answered the Virtual Reality Sickness Questionnaire (VRSQ) in order to determine if they felt any dizziness or discomfort due to movement within the virtual environment [23]. The questionnaires and their results can be seen in the following sections.

### 2.4. Conclusions

The design process results, including what has been learned and what still needs to be examined, will be addressed in the following sections of the paper.

### 3. Results and Conclusions

In this section, the results will be discussed, showing how users interact with the software. Afterward, in the Discussion section, several key points will be drawn from the development process.

As shown in the previous sections, software has been developed with the design and creation methodology, and it was tested with users. This process is iterative and is made up of several cycles in which each output is the first input for the next cycle. These cycles are short in order to minimize problems of design or programming. Traditional testing was also performed, such as black box and white box testing of the software, and the first release was made considering all the improvements and conclusions of the usability testing.

The main experience can be divided in four different sections: (1) tutorial and introduction scene; (2) collections of 3D pieces; (3) VR casual games and (4) information about local physical museums.

### 3.1. Tutorial and Introduction Scene

As mentioned in the design of the tutorial scene, the main purpose is to teach and show users how to interact with the software. The VR experience starts in a black environment and the tone of the colors slightly changes to dark brown (to avoid eye strain).

An orb appears with purple particles that spark in front of the user, see Figure 3a. Three-dimensional environment sounds were also developed in Unity to enrich the environment and interactions with the objects. This allows that every time a user loses focus on the orb, a sound is produced to hold their attention. After a couple of interactions with the orb, Maulina appears with her story, see Figure 3b.

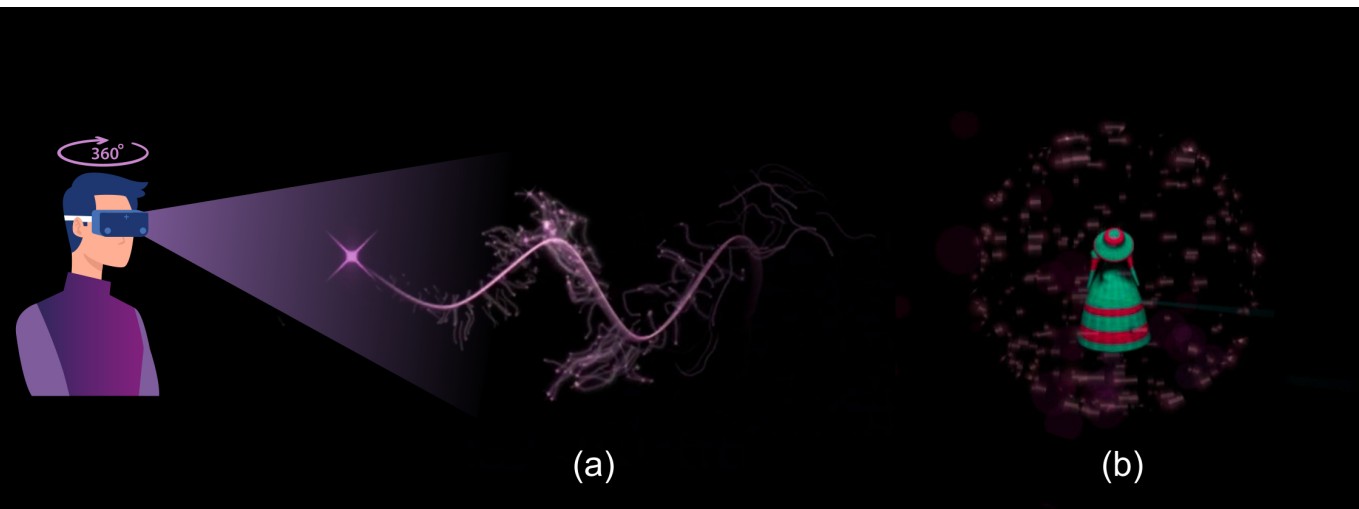

**Figure 3.** Introduction scene: (**a**) Orb that appears with purple particles that spark in front of the user. The orb moves to the next position once the user points at the orb with the controller and presses an action button. (**b**) Introduction of Maulina after the movement of the orb through different positions.

Maulina transforms the dark room into a magic museum environment that takes references from old and colonial architecture. Maulina prompts the user to point to a magic chest to open and discover the historical pieces that are contained within.

### 3.2. Collections of 3D Pieces

Once the user opens the chest, a front panel is created that prompts users to look at four collections: (1) Pre-columbian cultures; (2) Colony, crafts and rurality; (3) Chilean Independence; (4) Pedro Olmos [39], see Figure 3. Each collection of 3D pieces is grouped by the time period they represent. These pieces were obtained by the digitization process explained in the methodology section, with photogrammetry and 3D scanner techniques.

The collection of Pedro Olmos was digitized through high definition photography as the paintings are 2D objects.

Once the user chooses a collection, the front panel will show the pieces that are available to check out. Then, the user can select one piece with the controller. At this moment, the software shows users an interface where they can appreciate the 3D object in 360 degrees, access information, such as which museum the piece is from, and learn cultural and general information, see Figure 4.

The interface also allows the user to zoom in and examine more details, see Figure 5.

It is important to note that the experience was designed to give users the space to choose what they want to see and to set their own pace. No highlights were included to avoid holding or pointing their attention to any single piece or collection.

### 3.3. VR Casual Games

Three games were developed: (1) 2D sliding puzzle (piece restoration); (2) discover the historical 3D piece (excavation); and (3) find the differences, see Figure 6.

We included these games as a gamification part of the main VR experience, as the VR experience is not a game per se or would be considered a serious game. These games were not chosen at random but rather loosely inspired by aspects of the work that could go into finding and restoring a historical piece. Although the games are in no way real simulations of this work, the titles invite users (especially young users) to think about the jobs of piece restoration and conservation, excavation, and the role of observation in identifying pieces, as well as bolster their interest in the real museum pieces included in these games.

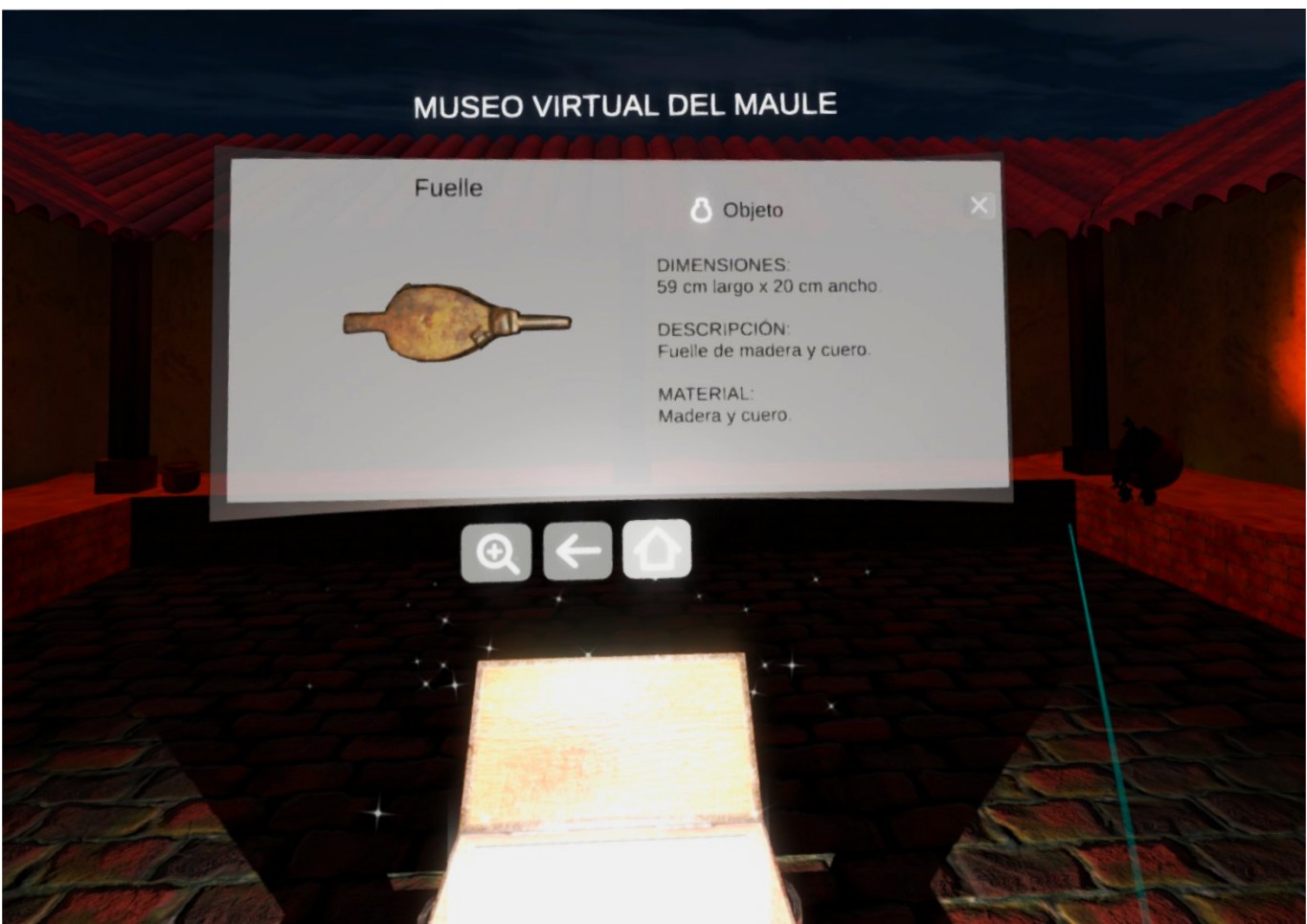

**Figure 4.** The panel of visualization: Interface where users can appreciate the 3D objects in 360 degrees and see all their features. In this case, a bellows with its dimensions, description, and material.

- 2D puzzle (Piece restoration): The puzzle is a grid of $3 \times 3$ with nine squares in total with a picture of a 3D piece. Once the game starts, each square moves randomly on the grid, disordering the original image. The challenge is to find the solution to the puzzle by sliding the squares on the grid. Since sometimes it is complicated for users to find a solution, a hint button was implemented. This button helps the user by showing the best movement in order to solve the sliding puzzle. This button was implemented with an iterative deepening A* algorithm, see Figure 6a.
- Discover the historical 3D piece (excavation): This game is a variation of the game Brick Breaker, in which users move a paddle to bounce a ball up towards bricks that break upon impact. Three pieces are hidden behind the bricks, which users need to uncover by breaking the bricks in front of them, as instructed by the panel, see Figure 6b.
- Find the differences: Two pictures are shown side by side on the GUI with an ad hoc environment. In one corner, the shapes of five pieces can be seen. The user needs to find the pieces and the differences between the two images. Once a piece is discovered, the shapes are replaced with the original piece, see Figure 6c.

Figure 5 image shows the Museo Virtual del Maule visualization panel.

**Figure 5.** Visualization panel: Zooming in on the bellows (piece from Museo Histórico de Villa Alegre); in the interface, the user can select auto spin or zoom in or out and go back to the original presentation (this picture was taken with the screenshot functionality of Oculus Quest).

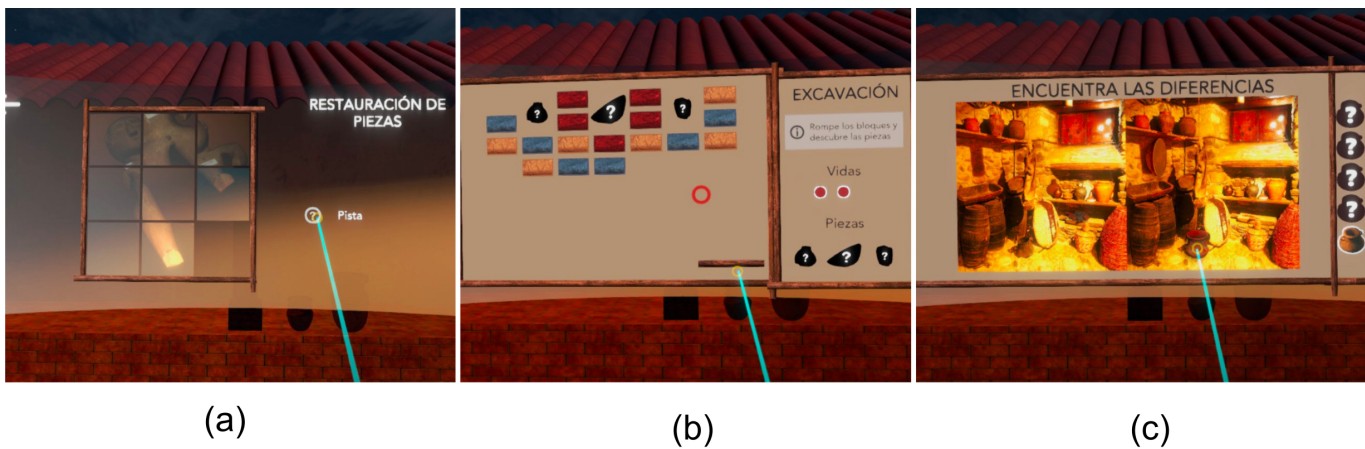

(a)  (b)  (c)

**Figure 6.** Games developed: (**a**) a 2D sliding puzzle (piece restoration); (**b**) discover the historical 3D piece (excavation); (**c**) find the differences.

All these games can be played on the right panel of the GUI with the controller, although each game is different in terms of complexity (low to medium). It is important to note that the VR experience is made for a wide range of users; therefore, the games implemented were casual and could be played with a low requirement of movement and minimal instructions.

### 3.4. Information of Physical Museums

Since the VR experience contains 3D pieces from nine museums from the Maule region in Chile, it is important to show users where the pieces can be found. In this context, we show this information every time users are interacting with the 3D piece in the front GUI. A panel was also implemented on the left with its own GUI. On this panel, statistics are shown, such as how many pieces are in each collection and how many museums are represented by these pieces.

There are two important moments in the experience when a whole map of the Maule region is shown, indicating the locations of the physical museums from which the 3D pieces are drawn. The first time is at the beginning, with the introduction of Maulina. At this point, Maulina prompts the user to look at the map and the locations of the museums, inviting them to visit these places in the future. The second time is when the user finishes the experience; Maulina thanks the users for visiting the VR museum and reminds them that they can see these pieces in person in the museums. The information included on the map is: the name of the region and area, as well as the street address of each museum.

Thus, if users participated in the VR experience with their private HDM in a space such as their home, or in a public place other than the museum (such as a school), then they can see where the pieces belong and learn about museums that they may not know.

### 3.5. Results of the Usability Study

All participants reacted positively while exploring the museum, and they were surprised and interested in learning about the entire virtual environment. One hundred percent of participants were able to explore the virtual museum at their own pace. No participant left the session without exploring most of the content.

As mentioned before, the objective of this study was to collect errors and problems in the software interface. A list of the most frequent errors found in the study is presented below in Table 3.

**Table 3.** Summary of the most relevant errors found in the study: The first column contains comments from the users about negative aspects. Although more comments were gathered, a selection of eight was made, which were the most important errors identified (most repeated, and which had the greatest impact on user experience). The second column has the solutions implemented in the final version.

| Negative Aspects Mentioned | Implemented Solution |
| --- | --- |
| Some texts were very long and could not be read well. | The texts were revised, the resolution was improved and the blurring of the letters was removed. |
| The buttons are small. | Buttons were enlarged, based on the design guidelines for interface development. |
| There is no way to go back to the first menu. | Go back buttons were added. |
| At the beginning, it is not known what to do, since I did not receive instructions and I thought it was loading the application. | The tutorial at the beginning of the experience was improved; it explains how to interact with the virtual environment. |
| When the chest is opened, it is not very easy to notice that there are panels on each side. | Signs were added to indicate the position of the panels to the user. |
| There is no change of soundtracks in the mini games. | Different soundtracks for the games were included. |
| Some of the buttons are misplaced. | The position of all the buttons was checked and they were positioned correctly. |

Although the purpose of the usability study is to identify any usability problems, the positive aspects of the virtual museum were also collected in order to identify what the user liked and to be able to enhance it to improve the user experience. A list of the salient positive aspects is presented below in Table 4.

**Table 4.** Summary of the most relevant positive aspects found in the study. Once again, although more comments were gathered, a selection of eight was made because they highlight different aspects of the virtual reality (VR) experience.

| Positive Aspects Mentioned |
| --- |
| Graphically it is beautiful and pleasing to the eye, the place is thematic and immersive. |
| It delivers information in a "nice" way, not over-reporting, but rather highlights the most important points. |
| The museum allows learning in a fun way. |
| Good introduction. |
| The music is immersive. |
| I was able to learn to use it easily. |
| Variety of content. |

*3.6. Presence Questionnaire*

This questionnaire consists of 24 questions related to realism, possibility to act and examine, quality of interface, and auto-evaluation of performance [38], see Table A1 (Appendix A). The participants marked the option that best represented them according to the content of the question and the descriptive labels of each question. In addition to answering the questionnaire, participants were asked to comment on the dimensions addressed in it. Next, general insights obtained from the qualitative analysis of the participants' responses will be presented.

- **Realism**: Overall, the opinion about this aspect was positive. Most of the participants stated that they found it to be a very realistic experience (88%), consistent with reality in terms of feeling like they are inside a museum. It was mentioned that the interaction was not completely natural, since the user had to interact with objects and elements through sight with the headset, which is quite different from how people interact with the environment in the real world. The majority of participants stated that they felt immersed inside the museum house, but aware that they were in a virtual reality experience. According to the participants, the control mechanisms within the environment were fairly real and they felt really involved in the virtual environment with all their senses.
- **Possibility to act and examine**: Regarding this aspect, the participants expressed they were not able to control everything within the virtual environment, since they could not correctly manipulate all the existing objects due to the limitation of only being able to use the head to move the raycast for selection. Despite this, the participants stated that they were able to examine the pieces and read the explanatory texts about them.
- **Quality of interface**: In general, the evaluation of the aesthetics of the interface was good, being one of the positive aspects most highlighted by participants. Most of the participants stated they could perform the tasks correctly without being distracted by the interface. According to participants, the control devices interfered somewhat with achieving their goals within the experience, due to what they expressed previously about only being able to control movement with the VR headset.
  There were no reports about having experienced a delay between the actions and the expected results, which means the feedback was shown immediately when they performed an action.

- **Auto-evaluation of performance**: Participants expressed that they were able to adapt quickly to the virtual environment and they could perform the tasks correctly despite only being able to use the headset for interactions.

### 3.7. Virtual Reality Sickness Questionnaire (VRSQ)

VRSQ measures motion sickness within a virtual environment and considers oculomotor and disorientation factors [38]. The VRSQ consists of asking which of the nine symptoms the user might feel when interacting with the virtual environment. The symptoms by factors are:

- Oculomotor: general discomfort, fatigue, eye strain, difficulty focusing.
- Disorientation: headache, fullness of head, blurred vision, dizzy (eyes closed), vertigo.

Participants were only asked which of the symptoms described in the VRSQ they felt; they were not asked about the severity of the symptoms. Of the total number of participants (n = 33), only three declared they had presented some slight discomfort when using the VR glasses. The symptoms reported were: blurred vision (n = 1), eye strain (n = 1) and difficulty in focusing (n = 1). All participants stopped feeling the discomfort when the virtual experience ended.

Some changes can be made in software development to minimize motion sickness. One of the most important is to reduce latency [40], that is, the time it takes for the movement made by the user to register in the software (the delay). If the time is too long, it can send the wrong signals to the brain, causing motion sickness, since the actual movement does not match what the user sees and hears. A delay of 20 ms is considered acceptable for some authors [41], which means that people can feel comfortable with a delay of 20 milliseconds. It is expected that, as virtual reality systems develop, the speed of latency will decrease, and it will significantly minimize the problem of motion sickness in VR experiences.

## 4. Discussion

This research aimed to design and develop a virtual museum with virtual reality technology that belongs to the local context with both intangible and tangible cultural aspects focused on the Maule region of Chile. The main contributions of this work are: (1) what has been learned and can be recommended based on the present experience of using VR technology to create a contextualized virtual museum, combining objects from multiple physical collections into one virtual collection that are united into a central narrative guided by a character drawn from local heritage; (2) the discussion about the experience of using a collaborative, iterative methodology resulting in software that invites users to know, visit and live the experience, the five steps of which are described with examples from the current work.

With respect to the first point, some recommendations can be offered on how to develop a culturally contextualized virtual museum. The present work suggests that this type of application could be used to help disseminate information about local museums, promote them and encourage users to visit them; therefore, an important recommendation is to include several actors in the development process from museums (such as curators) and target users. In the current work, a prototype was made and shared with these actors, allowing refinement of the prototype for future iterations. Another recommendation in the creation of a virtual museum is to allow users to enjoy the VR experience at their own pace. It is important to note that a full guided experience was purposefully avoided because users need to choose what to see and for how long, which details are important to them, and if they want to access to more information, and these are the types of behaviors that can be seen in a real visit to a place of this kind. Furthermore, it is recommended for a virtual museum to enrich the interactive feature that can not be obtained in other experiences, such as videos or online shows where users just see what is shown for them. An additional recommendation is that the resolution of the digitized pieces needs to be correlated with the HMD that will be used for viewing them. As stated previously, the VR experience was

developed for Oculus HMD devices. The digitization process was done with a framework that utilizes photogrammetry and a 3D scanner. It was important to consider which devices end-users will use to view the pieces and balance the resolution accordingly to optimize how they are visualized.

With respect to the second point, a methodology was followed that allowed the development of a VR experience considering users, creativity, digitization and software development. The following are some comments about the experience of using the design and creation strategy.

As described in the methodology section, the design and creation strategy consists of five steps: awareness, suggestion, development, evaluation and conclusion. Although traditional methodologies of development of software are focused on the end-user, in this study, end-users were also included in the creation process. Some key elements are important to note from this process. For the development of a VR experience, creativity is needed, just like the creation of video games (or other interactive software) requires a creative process, to make a useful and unique product for the end-users. This is where the awareness and suggestion steps of the design and creation strategy play a major role. This software product does not come from a client, but rather from a convergence of visions and discussions on the need of local museums to disseminate information using new technologies, and it required creativity to design a tentative idea of how to address this identified need and opportunity.

In the development step of the design and creation strategy, the VR experience was developed based on the creation of prototypes in an iterative process, which allowed quick iterations, testing features and improvements to the experience in each cycle. Some of the main advantages of this software that grew out of this process are: (1) it is an attractive and new technology for a wide audience, especially for young people; (2) the experience has a character to make a connection with the environment and let the user decide how and what to explore at their own pace; (3) it has functionalities for visualization of 3D pieces that are innovative and are the product of a usability test and users' opinions; (4) it has good definition balance of 3D pieces and optimization of the models for the VR experience; and (5) games are part of the experience for different audiences. Finally, it is worth stating that the addition of new functionalities in this VR experience are technically straightforward to implement because the software was made with Unity 3D with modular architecture and components.

In the evaluation step, the usability testing and analysis showed how important it is to include users in the developing process as it allows the design team and developers to identify issues before fully coding. In this way, it is easier to find errors and problems of interaction in the early stages of the development process, in order to improve the product and consider what the end-user needs to have an experience without UX issues. Therefore, it is less expensive to make changes in terms of the time required for the new development and its impact on the final result. In addition, getting feedback from the VRQS was another important process to include; because the VR technology is relatively new for the end-user, it is important to test and modify some designs or interactions that could cause some of the reported symptoms. The VRQS showed that for a majority of the users, the experience does not produce any of the described problematic symptoms (the symptoms only appeared in special cases, as was previously described), which means that a good user experience has been delivered. With the digitizing process and a correct balance of the number of polygons and textures of the 3D objects, the experience does not present lags in its use and display of programmed interactions.

Further work should be centered not only on ways to improve the software–user interaction but also to evaluate the impact of the virtual museum on users' attitudes toward the virtual experience, learning about cultural heritage pieces, or visiting a real museum. A better understanding of how the users' attitudes are formed or change as a result of an experience in a virtual museum would allow the development of better experiences for them and, potentially, a higher sense of community and belonging.

Similarly, another area of interest for future work is to compare differences between the type of technology in which the software can be used on users' attitudes and intentions to visit a real museum. Since the 3D objects have already been digitized, multiple different experiences could be developed, such as a typical visualization on a computer (on a web browser, for example, Sketchfab), a VR experience, a PC experience, or an interactive totem. Knowing this is crucial not only for technological reasons but also for making accessible the content of cultural heritage to different groups of the population that might vary in their socio-economic backgrounds.

**Author Contributions:** Conceptualization, F.B., L.J., I.G.; investigation, F.B. and I.G.; methodology, F.B., L.J.; writing—original draft, F.B. and L.J.; writing—review and editing, F.B., L.J. and I.G. All authors have read and agreed to the published version of the manuscript.

**Funding:** This research was funded by Gobierno Regional del Maule grant "fondo de innovación para la competitividad regional" number BIP 40001081.

**Institutional Review Board Statement:** Not applicable.

**Informed Consent Statement:** Not applicable.

**Data Availability Statement:** Data sharing is not applicable to this article.

**Acknowledgments:** We would like to thank the network of regional museums and their directors for having been open to participating in this work with a technological approach, as well as the Mauletec laboratory. We would also like to thank the Gobierno Regional del Maule for supporting the development of the VR virtual museum. Thank you to the University of Talca and the cultural extension department for their support.

**Conflicts of Interest:** The authors declare no conflict of interest. The funders had no role in the design of the study; in the collection, analyses, or interpretation of data; in the writing of the manuscript, or in the decision to publish the results.

## Abbreviations

The following abbreviations are used in this manuscript:

| | |
|---|---|
| VR | Virtual Reality |
| HDM | Head Mounted Display |
| GUI | Graphical User Interface |
| VE | Virtual Environment |

## Appendix A

**Table A1.** Questions of Witmer and Singer 24-question Presence Questionnaire.

| Questions |
|---|
| How much were you able to control events? |
| How responsive was the environment to actions that you initiated (or performed)? |
| How natural did your interactions with the environment seem? |
| How much did the visual aspects of the environment involve you? |
| How natural was the mechanism which controlled movement through the environment? |
| How compelling was your sense of objects moving through space? |
| How much did your experiences in the virtual environment seem consistent with your real world experiences? |
| Were you able to anticipate what would happen next in response to the actions that you performed? |
| How completely were you able to actively survey or search the environment using vision? |

**Table A1.** *Cont.*

| Questions |
| --- |
| How compelling was your sense of moving around inside the virtual environment? |
| How closely were you able to examine objects? |
| How well could you examine objects from multiple viewpoints? |
| How involved were you in the virtual environment experience? |
| How much delay did you experience between your actions and expected outcomes? |
| How quickly did you adjust to the virtual environment experience? |
| How proficient in moving and interacting with the virtual environment did you feel at the end of the experience? |
| How much did the visual display quality interfere or distract you from performing assigned tasks or required activities? |
| How much did the control devices interfere with the performance of assigned tasks or with other activities? |
| How well could you concentrate on the assigned tasks or required activities rather than on the mechanisms used to perform those tasks or activities? |
| How much did the auditory aspects of the environment involve you? |
| How well could you identify sounds? |
| How well could you localize sounds? |
| How well could you actively survey or search the virtual environment using touch? |
| How well could you move or manipulate objects in the virtual environment? |

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
