# Peer review of "Developing a Virtual Museum: Experience from the Design and Creation Process"

_information, doi:10.3390/info12060244_

Round 1

Reviewer 1 Report

The paper is original, the content is significant for the Journal, the quality and scientific soundness is appropriate. In my opinion, readers will find the paper of interest.

References are adequate and uptodate. 

Author Response

Thanks, I really appreciate your comments.

Reviewer 2 Report

This article suggests a current and attractive topic for the academy. The research is timely and worthwhile. The research problem is clearly defined. The authors provide fresh insight into the field.

The work structure is excellent and well-articulated. The literature review is detailed and thorough. To review the scientific sources from the industry can offer work of his colleagues: Mastykash, O., Peleshchyshyn, A., Fedushko, S., Trach, O. and Syerov, Y.: Internet Social Environmental Platforms Data Representation. 13th International Scientific and Technical Conference on Computer Sciences and Information Technologies (CSIT), Lviv, Ukraine, 2018, pp. 199-202. doi: 10.1109/STC-CSIT.2018.8526586

The results are explained in a clear and detailed manner.

Congratulations on a job well done.

Author Response

(The authors gave the same response as above.)

Reviewer 3 Report

The paper discusses the development of a virtual museum without proposing any novelty to state of the art and detailing the development process with a lot of knowledge that is trivial for researchers and practitioners in this specific field. As such, the article is not suited for publication in a scientific journal

Author Response

Thank you for your time and your comments.

Reviewer 4 Report

In this article authors designed and developed new simulated or contextualized experiences for a virtual museum. Furthermore, authors attempted use this technology to design a virtual museum that belongs to the local region with both intangible and tangible cultural aspects; thus creating a unique experience for users. They digitized 30 pieces from local museums with an experience guided by a character. A usability test was conducted. The participants reported feelings of high level of immersion and also realism. In addition, their experiment results showed that it is important to involve users in design and development (“creation” term used by the authors) process.

This article describes a research project that is well designed, created and implemented. All the required sections are presented with comprehensive details. The Introduction, Methodology, Results and Discussions sections are clearly presented and directly related to the objective of this research project. Overall, it is an excellent article with interesting topic and use of VR. Please see the following minor recommendations.

Recommendations:

  • Second paragraph in the “Introduction” section (lines 24 thru 29) is an important statement using “presence”, however, it is missing solid references. Please include at least the following and more.

North, M. M., & North, S. M. (2019). Dynamic immersive visualisation environments: enhancing pedagogical techniques. Australasian Journal of Information Systems23.

  • Please change subheading on line 87 from “Materials and Methods” to simply “Methodology” or “Methodology, Design and Creation”… Just remove the word “Materials” out of this subheading.

  • It is a good idea to either have a separate section for “4. Conclusions” or change “3. Results” on line 354 to “3. Results and Conclusions”.

Author Response

Thanks, I really appreciate your comments. With respect to the mentioned Recommendations:

1.- Two references were added for the paragraph and the term presence in line 26.

2.- We include the suggested reference and added a sentence providing context "Some studies have even shown that and immersive visualization 29
environment can help enhance learning" in line 29 -30.

3.- We changed subheading in line 88 from “Materials and Methods” to “Methodology”.

4.- We changed the subheading in line 355 from “3. Results”  to “3. Results and Conclusions”.

Please let us know any other recommendations. Thanks,